# In Vitro Dissolution Study of Acetylsalicylic Acid and Clopidogrel Bisulfate Solid Dispersions: Validation of the RP-HPLC Method for Simultaneous Analysis

**Ehlimana Osmanović Omerdić [1],*, Larisa Alagić-Džambić [2], Marko Krstić [3], Maja Pašić-Kulenović [1], Jadranka Odović [3] and Dragana Vasiljević [4]** 

[1] Development and Registration Department, Bosnalijek d.d., Jukićeva 53,
71000 Sarajevo, Bosnia and Herzegovina; maja.pasic-kulenovic@bosnalijek.com

[2] Quality Assurance and Quality Control Department, Bosnalijek d.d., Jukićeva 53,
71000 Sarajevo, Bosnia and Herzegovina; larisa.alagic-dzambic@bosnalijek.com

[3] Department of Analytical Chemistry, University of Belgrade—Faculty of Pharmacy, Vojvode Stepe 450,
11221 Belgrade, Serbia; marko.krstic@pharmacy.bg.ac.rs (M.K.); jadranka.odovic@pharmacy.bg.ac.rs (J.O.)

[4] Department of Pharmaceutical Technology and Cosmetology, University of Belgrade—Faculty of Pharmacy,
Vojvode Stepe 450, 11221 Belgrade, Serbia; dragana.vasiljevic@pharmacy.bg.ac.rs

* Correspondence: ehlimana.osmanovic.omerdic@bosnalijek.com; Tel.: +38761810619

**Abstract:** Solid dispersions were prepared via a solvent evaporation method, employing ethanol (96%, *v/v*) as solvent, with three different polymers as carrier: povidone, copovidone, and poloxamer 407. Previously developed reversed-phase HPLC (RP-HPLC) methods were modified and used for the simultaneous determination of acetylsalicylic acid and clopidogrel bisulfate and after release from solid dispersions. Chromatography was carried out on a C-18 column, with a mobile phase of acetonitrile–methanol–phosphate buffer pH 3.0, UV detection at 240 nm, and a run time of 6 min. The method was validated according to International Conference of Harmonisation guidelines and validation included specificity, accuracy, precision, linearity, robustness, limit of detection (LOD), and limit of quantification (LOQ). The method is specific for determination of acetylsalicylic acid and clopidogrel bisulfate. The linearity was provided in the concentration range 0.0275–0.1375 mg/mL for acetylsalicylic acid and 0.0200–0.1000 mg/mL for clopidogrel bisulfate, with a correlation coefficient ($R^2$ value) of 0.9999 for both active pharmaceutical ingredients (APIs). Accuracy was confirmed by calculated recoveries for acetylsalicylic acid (98.6–101.0%) and clopidogrel bisulfate (100.0–101.6%). The intra-day and the inter-day precision-calculated relative standard deviations are less than 1%, which indicates high precision of the method. The limits of detection and quantification for acetylsalicylic acid were 0.0004 and 0.0012 mg/mL, and for clopidogrel bisulfate 0.0002 mg/mL and 0.0007 mg/mL, respectively. Small variations in chromatographic conditions did not significantly affect qualitative and quantitative system responses, which proved robustness of method. The proposed RP-HPLC method was applied for simultaneous determination of clopidogrel bisulfate and acetylsalicylic acid from solid dispersions.

**Keywords:** clopidogrel bisulfate; acetylsalicylic acid; solid dispersion; dissolution; RP-HPLC; validation

## 1. Introduction

According to the World Health Organization (WHO), ischemic heart disease and stroke is the leading cause of adult-acquired disability and death worldwide [1]. Several antiplatelet drugs with different mechanisms of action are currently available for secondary prevention of cardiovascular

events such as ischemic stroke, myocardial infarction, and peripheral vascular disease. The most used oral antiplatelet drugs are acetylsalicylic acid, clopidogrel, dipyridamole, prasugrel, epoprostenol, and streptokinase. A combination of clopidogrel with acetylsalicylic acid may be substantially more effective than either drug alone [2–4]. Dual antiplatelet therapy appears to be safer and more effective in reducing stroke recurrence and combined vascular events in patients with acute ischemic stroke or transient ischemic attack, as compared with monotherapy [3,5]. In a large, randomized, blind, international trial (CURE), the safety and efficacy of the combination of clopidogrel and acetylsalicylic acid in patients with acute coronary syndromes were assessed. This study demonstrated that clopidogrel used with acetylsalicylic acid reduces death or myocardial infarction in patients undergoing intracoronary stent implantation by 77%, compared with acetylsalicylic acid alone [6]. Patient compliance has also shown improvement when a dosage form combining two active pharmaceutical ingredients (APIs) was used [7]. On the market, combination therapy that contains clopidogrel bisulfate and acetylsalicylic acid is available in the form of film-coated tablets.

Acetylsalicylic acid (ASA) (2-acetoxybenzoic acid) is a nonsteroidal anti-inflammatory drug and platelet aggregation inhibitor (Figure 1a). The mechanism of action is that of a cyclooxygenase inhibitor [8]. Acetylsalicylic acid is practically insoluble in water, and freely soluble in ethanol 96% *v/v* [9].

Clopidogrel bisulfate (CB) (S-methyl 2-(2-chlorophenyl-6,7-dihydrothieno [3,2-c]pyridine-5(4)-acetatesulfate) is a thienopiridine antiplatelet drug (Figure 1b). It is an antithrombotic agent that inhibits adenosine diphosphate-mediated platelet aggregation by selectively and irreversibly blocking platelet purinergic P2Y12 receptors [10,11]. Clopidogrel bisulfate is practically insoluble in water at neutral pH, but freely soluble at pH 1. It also dissolves freely in methanol, sparingly in methylene chloride, and is practically insoluble in ethyl ether [11].

(**a**)  (**b**)

**Figure 1.** Chemical structure of acetylsalicylic acid (**a**) and clopidogrel (**b**).

Development of formulation and validation of the dissolution method for poorly water soluble drugs has been a challenge in the pharmaceutical industry and for scientists, especially for drugs which belong to Biopharmaceutical Classification System (BCS) II or IV groups, like acetylsalicylic acid and clopidogrel bisulfate, with low solubility and/or permeability. Low solubility and dissolution rate are usually a limiting factor for oral drug absorption of these substances and consequently affect the bioavailability and therapeutic efficacy of the drug [12,13].

Formulations of solid dispersions (SD) were often used to increase the solubility and dissolution rate of low solubility drugs [12,14–17]. A solid dispersion is a dispersion of one or more active ingredients in a hydrophilic inert carrier matrix at solid state. Two basic procedures used to prepare solid dispersions are solvent evaporation and melting methods [15,16,18,19].

There are a number of HPLC methods available in the literature for the estimation of ASA and CB alone, and only a few methods are reported for simultaneous determination of both APIs [20–26], but there is no method reported for simultaneous estimation of ASA and CB after release from solid dispersion. This method was required primarily due to the application of excipients in solid dispersions

not used in previous studies to achieve adequate selectivity and also to shorten the retention time compared to the mentioned investigations.

The aim of this study was to modify previously developed [20,21] reversed-phase high-performance liquid chromatography (RP-HPLC) methods and to validate a new method for application in the simultaneous determination of ASA and CB released from solid dispersion. An in vitro dissolution method was conducted and validated according to United States Pharmacopeia (USP) and ICH guidelines [27,28]. The proposed RP-HPLC method should be applicable for routine quality control, stability testing, and estimating the concentrations of drugs in other solid dosage forms.

## 2. Materials and Methods

### 2.1. Materials

Acetylsalicylic acid, clopidogrel bisulfate, and excipients used for preparation of solid dispersion: povidone (Kollidon® 30, BASF, Ludwigshafen, Germany) copovidone (Kollidon® VA 64, BASF, Ludwigshafen, Germany) and poloxamer 407 (Kolliphor® P 407, BASF, Ludwigshafen, Germany) were pharmaceutical grade. Working standards, as well as reagents, were HPLC grade.

APIs, excipients, and other chemicals were a gift from Bosnalijek d.d. (Sarajevo, Bosnia and Herzegovina).

### 2.2. Methods

#### 2.2.1. Preparation of Solid Dispersions

Solid dispersions of acetylsalicylic acid and clopidogrel bisulfate with three different hydrophilic polymers (povidone, copovidone, and poloxamer 407) were prepared via the common solvent evaporation method, employing ethanol (96% *v/v*) as solvent. Solid dispersions were prepared in the of CB:ASA:carrier ratio, namely, 1:1.3:2.3, and injected into hard gelatinous capsules that contained a therapeutic dose of acetylsalicylic acid (100 mg) and clopidogrel (75 mg). The amount of polymer in the solid dispersions was chosen through earlier probe screening.

The required quantities of ASA, CB, and polymer were dissolved in the corresponding amount of solvent to get a clear solution. The solvent was then removed by evaporation under reduced pressure (vacuum) on a rotary vacuum evaporator (Rotavapor® R-205, Büchi, Switzerland) at a temperature of 55 °C with constant mixing. The mass obtained was crushed, pulverized, and shifted through sieve no. 500 (Ph. Eur.). Solid dispersions were injected into hard gelatinous capsules (size 0) so that they contained a therapeutic dose of ASA and CB.

#### 2.2.2. In Vitro Dissolution Method

In vitro dissolution method conditions were: Apparatus 1 (basket) speed of 75 rpm, in 900 mL of phosphate buffer (pH 6.8) at 37 °C ± 0.5 °C. Tests were conducted with a Vankel VK 7010 dissolution bath and a Varian fraction sample collection module. A sample volume of 1.5 mL was filtered (35 μm Micron Full Flow Filter, Agilent) and collected directly into an HPLC vial. The samples were collected at time points of 15, 30, 45, and 60 min. The test was also performed on empty hard gelatinous capsules to prove that they had no effect on the results obtained. Results were expressed as mean value ± standard deviation (S.D.) of three replicates.

Chromatographic Conditions

The final chromatographic conditions were RP-HPLC method, using a C-18 column; 4.6 mm × 150 mm; 5 μm maintained at 30 °C with a mobile phase of acetonitrile–methanol–phosphate buffer pH 3.0 (500:70:430 *v/v/v*). The mobile phase pumped through the column with a flow rate 1.2 mL/min, an injection volume of 20 μL and a run time of 6 min. Detection was carried out at 240 nm by a UV Detector.

Preparation of Standard Stock Solutions

A standard stock solution of ASA was prepared by dissolving 25 mg of ASA in phosphate buffer pH 6.8 (0.25 mg/mL for ASA). A standard stock solution of CB was prepared by dissolving 20 mg of CB in 1 mL of methanol and diluted with phosphate buffer pH 6.8 to 100 mL (0.20 mg/mL for CB). Subsequently, these stock solutions were used to prepare a series of two-component working standard solutions through dilution with phosphate buffer pH 6.8.

Validation of Method

The method was validated according to International Conference of Harmonisation guidelines and validation included specificity, linearity, accuracy, precision, limit of detection (LOD), limit of quantification (LOQ), and robustness.

Specificity was studied by injecting ASA and CB standard solutions, and solutions of polymers, in phosphate buffer pH 6.8 used in formulation studies. Specificity was also determined by injecting solution after disintegration of empty hard gelatinous capsules under the same RP-HPLC conditions to ensure that there was no interference. The chromatograms were examined at retention time of acetylsalicylic acid and clopidogrel.

The linearity of the method was tested in the range 0.0275–0.1375 mg/mL for ASA, and 0.0200–0.1000 mg/mL for CB (Table 1).

The accuracy was tested on three concentration levels: for ASA, 0.0550, 0.1100, and 0.1375 mg/mL; and for CB, 0.0400, 0.0800, and 0.1000 mg/mL (50%, 100% and 125%) of the target concentration (in this study, the target concentration (100%) was 0.1100 mg/mL for ASA and 0.0800 mg/mL for CB). Recovery (%) was calculated for each concentration and mean recovery value should be between 90% and 110% to be accepted.

One level of method precision (as system precision) was tested by analysis of ASA and CB at three concentration levels—50%, 100%, and 125%. The intra-day precision of the method was determined from replicate analysis (n = 5) of ASA and CB test standards at concentrations within the linear range of the assay for each drug (0.0550, 0.1100, and 0.1375 mg/mL for ASA and 0.0400, 0.0800, and 0.1000 mg/mL for CB). The inter-day precision of the method was determined by replicate analysis (n = 5) of ASA and CB with three different concentrations on two different days. The results were acceptable if the relative standard deviations (RSD) is less than 2% [28].

The values of LOD and LOQ were determined as described before by Alquadeib [29].

Robustness was evaluated by accompanying small variations in chromatographic conditions, such as alteration in the pH of the mobile phase, percentage organic content, changes in temperature, and the impact of these changes on qualitative and quantitative system responses.

## 3. Results and Discussion

An RP-HPLC method was utilized to provide adequate specificity and a short run time. The optimized chromatographic separation is shown in Figure 2. Chosen RP-HPLC conditions provided satisfactory separation of acetylsalicylic acid and clopidogrel from each other. All the system suitability parameters were optimized by freshly prepared standard solution of acetylsalicylic acid (0.1100 mg/mL) and clopidogrel (0.0800 mg/mL).

The method was validated for simultaneous estimation of acetylsalicylic acid and clopidogrel combinations using specificity, linearity, accuracy, precision, limit of detection (LOD), limit of quantification (LOQ), and robustness. Average retention time for acetylsalicylic acid and clopidogrel was 1.9 and 4 min, respectively, and was shorter than other published data [20–26]. Shorter retention time enables shorter analysis time, which is very important in routine analysis, especially in the pharmaceutical industry.

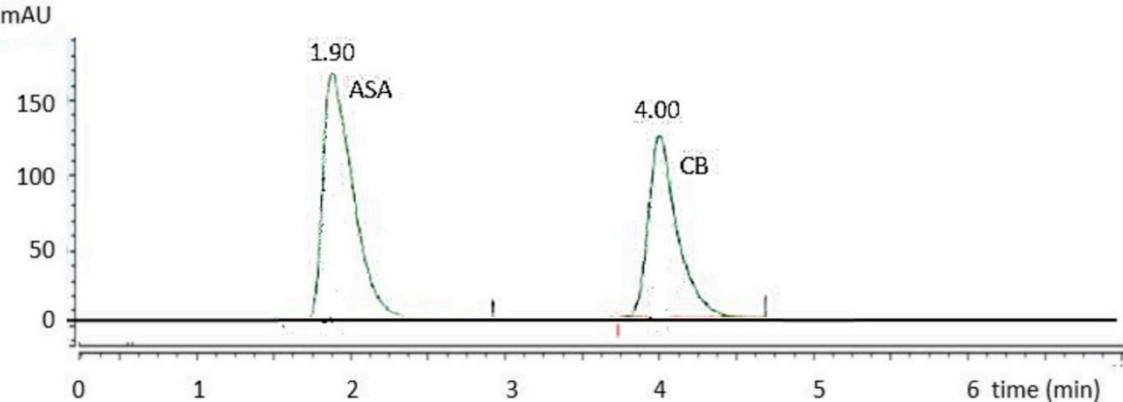

**Figure 2.** Representative chromatogram of acetylsalicylic acid (ASA) and clopidogrel (CB) mixed standard.

## 3.1. Specificity

The high degree of specificity of the method was confirmed by analyzing chromatograms obtained by injecting the placebo and standard solution. No peak was identified near ASA and CB retention times. A study to establish the interference of empty hard gelatinous capsules was conducted. Two peaks at retention times of 2.58 and 9.73 min appeared on the capsule chromatogram, so there was no overlap with active substance peaks (1.9 and 4 min) and no interactions, which implies selectivity of the method. The chromatogram of empty hard gelatinous capsules is shown in Figure 3.

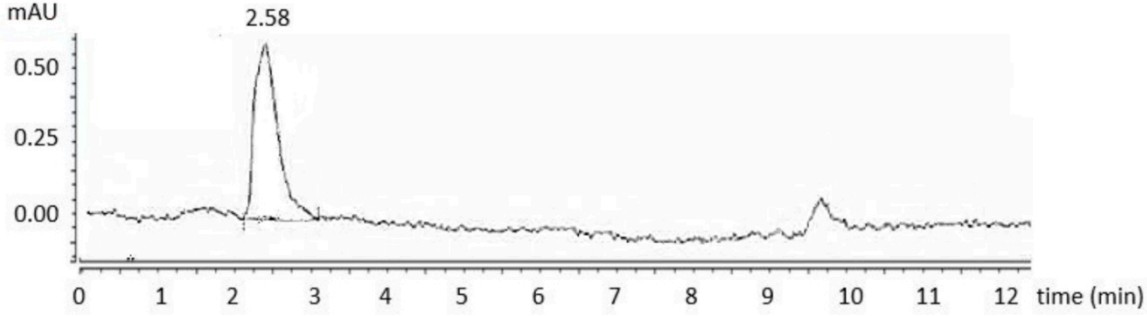

**Figure 3.** Representative chromatogram of empty hard gelatinous capsules.

## 3.2. Linearity

Linearity was evident in the concentration range 0.0275–0.1375 mg/mL for acetylsalicylic acid and 0.0200–0.1000 mg/mL for clopidogrel bisulfate with a correlation co-efficient ($R^2$ value) of 0.9999 (Table 1, Figures 4 and 5).

**Table 1.** Linearity results for acetylsalicylic acid and clopidogrel bisulfate.

| | Acetylsalicylic Acid | | | Clopidogrel Bisulfate | |
|---|---|---|---|---|---|
| D (%) | Concentration of Standard Solution (mg/mL) | Peak Area (mAU*s) | D (%) | Concentration of Standard Solution (mg/mL) | Peak Area (mAU*s) |
| 25 | 0.0275 | 558,812 | 25 | 0.0200 | 309,513 |
| 40 | 0.0440 | 888,893 | 40 | 0.0320 | 494,592 |
| 45 | 0.0495 | 1,001,648 | 45 | 0.0360 | 558,577 |
| 50 | 0.0550 | 1,118,005 | 50 | 0.0400 | 620,662 |
| 100 | 0.1100 | 2,191,888 | 100 | 0.0800 | 1,225,652 |
| 125 | 0.1375 | 2,727,058 | 125 | 0.1000 | 1,526,992 |
| $R^2$ | 0.9999 | | $R^2$ | 0.9999 | |

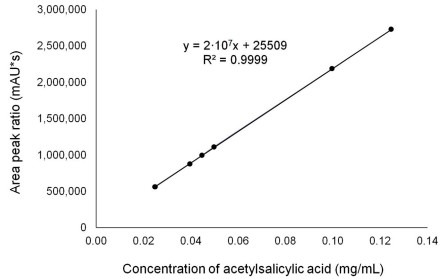

**Figure 4.** Linearity plot for acetylsalicylic acid.

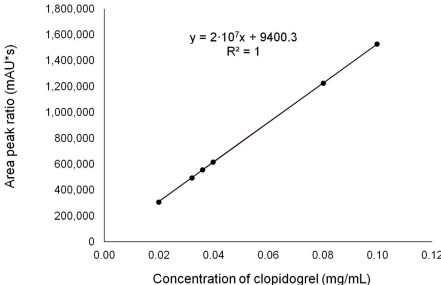

**Figure 5.** Linearity plot for clopidogrel.

*3.3. Accuracy*

The accuracy of the method was determined by recovery experiments. The accuracy studies were carried out at three concentrations: 0.0550, 0.1100, and 0.1375 mg/mL for ASA (50%, 100%, and 125%) and 0.0400, 0.0800, and 0.1000 mg/mL for CB (50%, 100%, and 125%). The percentage recovery was calculated from the data obtained. The results are shown in Tables 2 and 3. Mean recoveries of acetylsalicylic acid and clopidogrel were 98.6–101.0% and 100.0–101.6%, respectively, confirming the accuracy of the method.

**Table 2.** Accuracy results for acetylsalicylic acid.

| Level (%) | Concentration of Standard Solution (mg/mL) | Obtained Concentration (mg/mL) | Recovery(%) |
|---|---|---|---|
| 50 | 0.0550 | 0.0556 | 101.8 |
| 100 | 0.1100 | 0.1089 | 99.0 |
| 125 | 0.1375 | 0.1356 | 98.6 |
| Mean recovery (%) | | | 99.5 |

**Table 3.** Accuracy results for clopidogrel.

| Level (%) | Concentration of Standard Solution (mg/mL) | Obtained Concentration (mg/mL) | Recovery(%) |
|---|---|---|---|
| 50 | 0.0400 | 0.0406 | 101.6 |
| 100 | 0.0800 | 0.0802 | 100.3 |
| 125 | 0.1000 | 0.1000 | 100.0 |
| Mean recovery (%) | | | 100.6 |

*3.4. Precision*

System precision studies were evaluated by five replicate measurements at three concentrations (intra-day) and on two different days over a week (inter-day). The % RSD was found to be less than 1% for both substances, which indicates high precision of method (Tables 4 and 5). In practical analysis, that means that average results are statistically equivalent [30]. RSD values were lower than results shown in two other mentioned methods [20,21].

**Table 4.** Precision results for acetylsalicylic acid.

| Injections | Peak Area (mAU*s) Acetylsalicylic Acid | | | | | |
| | 50% 1. day | 50% 2. day | 100% 1. day | 100% 2. day | 125% 1. day | 125% 2. day |
| --- | --- | --- | --- | --- | --- | --- |
| 1 | 1,118,005 | 1,119,005 | 2,196,838 | 2,148,464 | 2,727,058 | 2,730,364 |
| 2 | 1,116,341 | 1,121,366 | 2,208,228 | 2,146,395 | 2,730,215 | 2,725,634 |
| 3 | 1,121,242 | 1,123,216 | 2,212,075 | 2,149,707 | 2,729,241 | 2,719,863 |
| 4 | 1,115,786 | 1,118,345 | 2,216,265 | 2,148,082 | 2,715,653 | 2,740,793 |
| 5 | 1,110,211 | 1,115,256 | 2,214,434 | 2,139,806 | 2,715,934 | 2,720,256 |
| Mean area peak (mAU*s) | 1,116,317 | 1,119,438 | 2,209,568 | 2,146,491 | 2,723,620 | 2,727,382 |
| RSD (%) | 0.36 | 0.29 | 0.35 | 0.18 | 0.27 | 0.32 |

**Table 5.** Precision results for clopidogrel bisulfate.

| Injections | Peak Area (mAU*s) Clopidogrel Bisulfate | | | | | |
| | 50% 1. day | 50% 2. day | 100% 1. day | 100% 2. day | 125% 1. day | 125% 2. day |
| --- | --- | --- | --- | --- | --- | --- |
| 1 | 620,662 | 621,251 | 1,220,200 | 1,227,386 | 1,526,992 | 1,525,632 |
| 2 | 619,432 | 619,534 | 1,225,194 | 1,215,347 | 1,519,965 | 1,518,649 |
| 3 | 619,578 | 620,097 | 1,219,874 | 1,238,342 | 1,528,761 | 1,526,489 |
| 4 | 621,392 | 619,987 | 1,218,465 | 1,237,379 | 1,523,235 | 1,517,398 |
| 5 | 620,978 | 621,872 | 1,224,411 | 1,210,359 | 1,527,691 | 1,527,821 |
| Mean area peak (mAU*s) | 620,408 | 620,548 | 1,221,629 | 1,227,563 | 1,525,329 | 1,523,198 |
| RSD (%) | 0.14 | 0.16 | 0.24 | 0.84 | 0.24 | 0.32 |

### 3.5. LOD and LOQ

The limits of detection and quantification were 0.0004 and 0.0012 mg/mL for ASA, and 0.0002 and 0.0007 mg/mL for CB, respectively.

### 3.6. Robustness

There were no changes in the chromatographic response for varied chromatographic parameters, thus confirming the robustness of the method.

### 3.7. In Vitro *Dissolution Data*

The development of the present RP-HPLC method has made it a possibility to prove that the selected polymers for solid dispersion preparations have an influence on the dissolution rate of poorly soluble drugs, like ASA and CB. The percentage of dissolved drugs from solid dispersions and pure APIs is presented in Table 6. All three prepared solid dispersions showed an improvement in the percentage of the released drug after 60 min for both active substances, compared to the pure APIs. The reason for such improvements could be because APIs in SD are surrounded by the molecules of the hydrophilic polymers, which improves wetting and dissolution [19,31,32].

After 60 min, in case of ASA, the SD containing povidone and poloxamer 407 had the highest improvement of dissolution rate (98.63% and 97.46%, respectively), in comparison with pure ASA (82.02%). For CB, the highest improvement of dissolution rate was for SD with poloxamer 407, 59.56% (32.20% for pure CB). SD with poloxamer 407 had the most significant results for both APIs (59.56% for CB and 97.46% for ASA). An increase in the ratio of polymers could have the effect of increasing the dissolution rate of CB, and promising results could be obtained with a SD using poloxamer 407.

**Table 6.** Percent (%) of dissolved drugs from solid dispersions and pure APIs.

| | Percent (%) of Dissolved Drugs * | | | | | | | |
|---|---|---|---|---|---|---|---|---|
| | | | Polymer in SD | | | | | |
| | Pure Drug | | Povidone | | Copovidone | | Poloxamer 407 | |
| API | ASA | CB | ASA | CB | ASA | CB | ASA | CB |
| Time (min) | 0 | 0 | 0 | 0 | 0 | 0 | 0 | 0 |
| 15 | 61.40 (0.21) | 26.81 (0.07) | 58.98 (0.93) | 27.87 (0.04) | 17.58 (0.56) | 10.38 (0.86) | 28.44 (0.46) | 19.71 (0.36) |
| 30 | 76.02 (0.50) | 28.05 (0.03) | 84.98 (0.52) | 38.90 (0.32) | 52.43 (0.48) | 25.20 (0.15) | 64.33 (0.33) | 39.71 (0.33) |
| 45 | 80.58 (0.28) | 32.07 (0.02) | 97.11 (0.54) | 39.03 (0.68) | 71.95 (0.55) | 34.65 (0.04) | 86.39 (0.30) | 54.12 (0.16) |
| 60 | 82.02 (0.08) | 32.20 (0.12) | 98.63 (0.77) | 39.65 (0.54) | 85.51 (0.64) | 39.68 (0.71) | 97.46 (0.37) | 59.56 (0.03) |

* Mean value (± standard deviation).

## 4. Conclusions

The proposed RP-HPLC method was validated for simultaneous estimation of acetylsalicylic acid and clopidogrel bisulfate. All the validation parameters for both active substances meet the criteria, according to ICH guidelines, which indicates the validity of the method. This analytical method is simple, selective, accurate, precise, robust, and linear. The method was successfully modified with a significantly shorter retention time, higher precision, high accuracy, and low detection and quantification limits. Because of the shorter retention time, this method shortens analysis time, which is very important in research, and also in the pharmaceutical industry.

This RP-HPLC method has been successfully used to determine acetylsalicylic acid and clopidogrel bisulfate after release from solid dispersions. All prepared solid dispersions showed improvement in the percent of dissolved drug, compared to pure APIs, but the best results for both APIs occurred in solid dispersions with poloxamer 407. In further research, the developed method would be very helpful in understanding how the chosen hydrophilic polymer ratio in solid dispersions affects the dissolution rate of ASA and CB.

**Author Contributions:** Conceptualization, E.O.O. and D.V.; methodology, E.O.O., M.K., L.A.-D., and D.V.; software J.O.; validation, L.A.-D., M.K., and J.O.; formal analysis, E.O.O., M.K., L.A.-D., J.O., and D.V.; investigation, E.O.O., M.K., and L.A.D.Ž.; resources, E.O.O., M.P.-K., and L.A.-D.; data curation, E.O.O., M.P.-K., and L.A.-D.; writing—original draft preparation, E.O.O., L.A.-D., and M.P.-K.; writing—review and editing, D.V. and J.O.; visualization, J.O. and D.V.; supervision, D.V.; project administration, M.P.-K.; funding acquisition, J.O., D.V., and M.P.-K. All authors have read and agreed to the published version of the manuscript.

**Funding:** This research received no external funding.

**Acknowledgments:** The authors gratefully acknowledge Bosnalijek d.d. and the Ministry of Education, Science and Technological Development of the Republic of Serbia for their support.

**Conflicts of Interest:** The authors declare no conflicts of interest.

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
