# Peer review of "In Vitro Dissolution Study of Acetylsalicylic Acid and Clopidogrel Bisulfate Solid Dispersions: Validation of the RP-HPLC Method for Simultaneous Analysis"

_applsci, doi:10.3390/app10144792_

Round 1

Reviewer 1 Report

The authors present an interested study.

Method validation is well presented and well executed, but i suggest the introduction of Mandel' s test as a further confirmation of a good linearity.

I recommend the publication.

Author Response

Thank you very much for very useful suggestion, positive feedback and recommendation for publishing our manuscript. We considered Mandel’ s test, but all our results in this paper were calculated according to presented calibrate curves.

Reviewer 2 Report

The manuscript by Omerdic et al. presents an analytical separation of a solid drug dispersion. As this is an important area of drug delivery, the paper warrants publication. A few minor edits may be in order:

  1. Lines 21-24 present too much detail compared to what is typically found in an abstract.
  2. Lines 80-83 should be the heart of why this project was performed and consequently need to be expanded. Just because no one has doe this separation before is not sufficient reason for the project. What is the rationale and what were the criteria established in developing the project?
  3. Lines 176-177 and 245, shortened retention time does not necessarily lead to improved efficiency. Efficiency is a function of the narrowness of a chromatographic peak. Either calculate the efficiency to show the rader that efficiency is improved, or delete this statement.

Author Response

Point 1: Lines 21-24 present too much detail compared to what is typically found in an abstract.

Response 1: Thank you for the suggestions that will contribute to the quality of the article. We made corrections and deleted some details.

The sentence (lines 21-24):

Chromatography was carried out on a C-18 column; 4.6 mm x 150 mm; 5 µm maintained at 30 °C with a mobile phase of acetonitrile–methanol–phosphate buffer pH 3.0 (500:70:430, v/v/v). Flow rate was 1.2 mL/min, UV detection at 240 nm, an injection volume of 20 μL, and a run time of 6 min.

was replaced with the following text:

Chromatography was carried out on a C-18 column with a mobile phase of acetonitrile–methanol–phosphate buffer pH 3.0, UV detection at 240 nm, and a run time of 6 min.

Point 2: Lines 80-83 should be the heart of why this project was performed and consequently need to be expanded. Just because no one has doe this separation before is not sufficient reason for the project. What is the rationale and what were the criteria established in developing the project?

Response 2: We expanded the explanation, why this project was performed, with the sentence (lines 89-91):

This method was required primarily due to the application of excipients in solid dispersions not used in previous studies to achieve adequate selectivity and also to shorten the retention time compared to the mentioned investigations.

Point 3: Lines 176-177 and 245, shortened retention time does not necessarily lead to improved efficiency. Efficiency is a function of the narrowness of a chromatographic peak. Either calculate the efficiency to show the rader that efficiency is improved, or delete this statement.

Response 3: We deleted the statement in the sentence lines 218-220 and 302:

Shorter retention time enables better method efficiency and shortens analysis time, which is a very important in routine analysis, especially in pharmaceutical industry.

and replaced with the following text:

Shorter retention time enables shortens analysis time, which is a very important in routine analysis, especially in pharmaceutical industry.